# LOCAL SGD CONVERGES FAST AND COMMUNICATES LITTLE

**Sebastian U. Stich**
EPFL, Switzerland
sebastian.stich@epfl.ch

## ABSTRACT

Mini-batch stochastic gradient descent (SGD) is state of the art in large scale distributed training. The scheme can reach a linear speedup with respect to the number of workers, but this is rarely seen in practice as the scheme often suffers from large network delays and bandwidth limits. To overcome this communication bottleneck recent works propose to reduce the communication frequency. An algorithm of this type is *local SGD* that runs SGD independently in parallel on different workers and averages the sequences only once in a while. This scheme shows promising results in practice, but eluded thorough theoretical analysis.

We prove concise convergence rates for local SGD on convex problems and show that it converges at the same rate as mini-batch SGD in terms of number of evaluated gradients, that is, the scheme achieves linear speedup in the number of workers and mini-batch size. The number of communication rounds can be reduced up to a factor of $T^{1/2}$—where $T$ denotes the number of total steps—compared to mini-batch SGD. This also holds for asynchronous implementations.

Local SGD can also be used for large scale training of deep learning models. The results shown here aim serving as a guideline to further explore the theoretical and practical aspects of local SGD in these applications.

## 1 INTRODUCTION

Stochastic Gradient Descent (SGD) (Robbins & Monro, 1951) consists of iterations of the form

$$\mathbf{x}_{t+1} := \mathbf{x}_t - \eta_t \mathbf{g}_t \,, \tag{1}$$

for iterates (weights) $\mathbf{x}_t, \mathbf{x}_{t+1} \in \mathbb{R}^d$, stepsize (learning rate) $\eta_t > 0$, and stochastic gradient $\mathbf{g}_t \in \mathbb{R}^d$ with the property $\mathbb{E}\,\mathbf{g}_t = \nabla f(\mathbf{x}_t)$, for a loss function $f \colon \mathbb{R}^d \to \mathbb{R}$. This scheme can easily be parallelized by replacing $\mathbf{g}_t$ in (1) by an average of stochastic gradients that are independently computed in parallel on separate workers (*parallel SGD*). This simple scheme has a major drawback: in each iteration the results of the computations on the workers have to be shared with the other workers to compute the next iterate $\mathbf{x}_{t+1}$. Communication has been reported to be a major bottleneck for many large scale deep learning applications, see e.g. (Seide et al., 2014; Alistarh et al., 2017; Zhang et al., 2017; Lin et al., 2018b). *Mini-batch* parallel SGD addresses this issue by increasing the compute to communication ratio. Each worker computes a mini-batch of size $b \geq 1$ before communication. This scheme is implemented in state-of-the-art distributed deep learning frameworks (Abadi et al., 2016; Paszke et al., 2017; Seide & Agarwal, 2016). Recent work in (You et al., 2017; Goyal et al., 2017) explores various limitations of this approach, as in general it is reported that performance degrades for too large mini-batch sizes (Keskar et al., 2016; Ma et al., 2018; Yin et al., 2018).

In this work we follow an orthogonal approach, still with the goal to increase the compute to communication ratio: Instead of increasing the mini-batch size, we reduce the communication frequency. Rather than keeping the sequences on different machines in sync, we allow them to evolve *locally* on each machine, independent from each other, and only average the sequences once in a while (*local SGD*). Such strategies have been explored widely in the literature, under various names.

An extreme instance of this concept is *one-shot SGD* (McDonald et al., 2009; Zinkevich et al., 2010) where the local sequences are only exchanged once, after the local runs have converged. Zhang

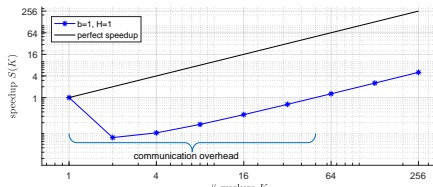 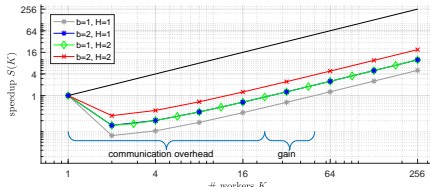

Figure 1: Illustration of the speedup (3) for time-to-accuracy when either increasing mini-batch size $b$ $(1 \rightarrow 2)$ or communication inverval $H$ $(1 \rightarrow 2)$, for compute to communication ratio $\rho = 25$.

et al. (2013) show statistical convergence (see also (Shamir & Srebro, 2014; Godichon-Baggioni & Saadane, 2017; Jain et al., 2018)), but the analysis restricts the algorithm to at most one pass over the data, which is in general not enough for the training error to converge. More practical are schemes that perform more frequent averaging of the parallel sequences, as e.g. (McDonald et al., 2010) for perceptron training (*iterative parameter mixing*), see also (Coppola, 2015), (Zhang et al., 2014; Bijral et al., 2016; Zhang et al., 2016) for the training of deep neural networks (*model averaging*) or in federated learning (McMahan et al., 2017).

The question of how often communication rounds need to be initiated has eluded a concise theoretical answer so far. Whilst there is practical evidence, the theory does not even resolve the question whether averaging helps when optimizing convex functions. Concretely, whether running local SGD on $K$ workers is $K$ times faster than running just a single instance of SGD on one worker.[1]

We fill this gap in the literature and provide a concise convergence analysis of local SGD. We show that averaging helps. Frequent synchronization of $K$ local sequences increases the convergence rate by a factor of $K$, i.e. a linear speedup can be attained. Thus, local SGD is as efficient as parallel mini-batch SGD in terms of computation, but the communication cost can be drastically reduced.

## 1.1 CONTRIBUTIONS

We consider finite-sum convex optimization problems $f \colon \mathbb{R}^d \to \mathbb{R}$ of the form

$$f(\mathbf{x}) = \frac{1}{n} \sum_{i=1}^{n} f_i(\mathbf{x}), \qquad \mathbf{x}^* := \arg\min_{\mathbf{x} \in \mathbb{R}^d} f(\mathbf{x}), \qquad f^\star := f(\mathbf{x}^\star), \qquad (2)$$

where $f$ is $L$-smooth[2] and $\mu$-strongly convex[3]. We consider $K$ parallel mini-batch SGD sequences with mini-batch size $b$ that are synchronized (by averaging) after at most every $H$ iterations. For appropriate chosen stepsizes and an averaged iterate $\hat{\mathbf{x}}_T$ after $T$ steps (for $T$ sufficiently large, see Section 3 below for the precise statement of the convergence result with bias and variance terms) and synchronization delay $H = O(\sqrt{T/(Kb)})$ we show convergence

$$\mathbb{E} f(\hat{\mathbf{x}}_T) - f^\star = O\left(\frac{G^2}{\mu b K T}\right), \qquad (3)$$

with second moment bound $G^2 \geq \mathbb{E}\|\nabla f_i(\mathbf{x})\|^2$. Thus, we see that compared to parallel mini-batch SGD the communication rounds can be reduced by a factor $H = O(\sqrt{T/(Kb)})$ without hampering the asymptotic convergence. Equation (3) shows perfect linear speedup in terms of computation, but with much less communication that mini-batch SGD. The resulting speedup when taking communication cost into account is illustrated in Figure 1 (see also Section D below). Under the assumption that (3) is tight, one has thus now two strategies to improve the compute to communication ratio (denoted by $\rho$): (i) either to increase the mini-batch size $b$ or (ii) to increase the communication interval $H$. Both strategies give the same improvement when $b$ and $H$ are small (linear speedup). Like mini-batch SGD that faces some limitations for $b \gg 1$ (as discussed in e.g. (Dekel et al., 2012; Ma et al., 2018; Yin et al., 2018)), the parameter $H$ cannot be chosen too large in local SGD. We give some pratical guidelines in Section 4.

Our proof is simple and straightforward, and we imagine that—with slight modifications of the proof—the technique can also be used to analyze other variants of SGD that evolve sequences on

---

[1]On convex functions, the average of the $K$ local solutions can of course only decrease the objective value, but convexity does not imply that the averaged point is $K$ times better.

[2]$f(\mathbf{y}) \leq f(\mathbf{x}) + \langle \nabla f(\mathbf{x}), \mathbf{y} - \mathbf{x} \rangle + \frac{L}{2} \|\mathbf{y} - \mathbf{x}\|^2, \forall \mathbf{x}, \mathbf{y} \in \mathbb{R}^d$.

[3]$f(\mathbf{y}) \geq f(\mathbf{x}) + \langle \nabla f(\mathbf{x}), \mathbf{y} - \mathbf{x} \rangle + \frac{\mu}{2} \|\mathbf{y} - \mathbf{x}\|^2, \forall \mathbf{x}, \mathbf{y} \in \mathbb{R}^d$.

different worker that are not perfectly synchronized. Although we do not yet provide convergence guarantees for the non-convex setting, we feel that the positive results presented here will spark further investigation of local SGD for this important application (see e.g. (Yu et al., 2018)).

## 1.2 RELATED WORK

A parallel line of work reduces the communication cost by compressing the stochastic gradients before communication. For instance, by limiting the number of bits in the floating point representation (Gupta et al., 2015; Na et al., 2017; Sa et al., 2015), or random quantization (Alistarh et al., 2017; Wen et al., 2017). The ZipML framework applies this technique also to the data (Zhang et al., 2017). Sparsification methods reduce the number of non-zero entries in the stochastic gradient (Alistarh et al., 2017; Wangni et al., 2017). A very aggressive—and promising—sparsification method is to keep only very few coordinates of the stochastic gradient by considering only the coordinates with the largest magnitudes (Seide et al., 2014; Strom, 2015; Dryden et al., 2016; Aji & Heafield, 2017; Sun et al., 2017; Lin et al., 2018b; Stich et al., 2018).

Allowing asynchronous updates provides an alternative solution to disguise the communication overhead to a certain amount (Niu et al., 2011; Sa et al., 2015; Lian et al., 2015), though alternative strategies might be better when high accuracy is desired (Chen et al., 2016). The analysis of Agarwal & Duchi (2011) shows that asynchronous SGD on convex functions can tolerated delays up to $O(\sqrt{T/K})$, which is identical to the maximal length of the local sequences in local SGD. Asynchronous SGD converges also for larger delays (see also (Zhou et al., 2018)) but without linear speedup, a similar statement holds for local SGD (see discussion in Section 3). The current frameworks for the analysis of asynchronous SGD do not cover local SGD. A fundamental difference is that asynchronous SGD maintains a (almost) synchronized sequence and gradients are computed with respect this unique sequence (but just applied with delays), whereas each worker in local SGD evolves a different sequence and computes gradient with respect those iterates.

For the training of deep neural networks, Bijral et al. (2016) discuss a stochastic averaging schedule whereas Zhang et al. (2016) study local SGD with more frequent communication at the beginning of the optimization process. The elastic averaging technique (Zhang et al., 2015) is different to local SGD, as it uses the average of the iterates only to guide the local sequences but does not perform a hard reset after averaging. Among the first theoretical studies of local SGD in the non-convex setting are (Coppola, 2015; Zhou & Cong, 2018) that did not establish a speedup, in contrast to two more recent analyses (Yu et al., 2018; Wang & Joshi, 2018). Yu et al. (2018) show linear speedup of local SGD on non-convex functions for $H = O(T^{1/4}K^{-3/4})$, which is more restrictive than the constraint on $H$ in the convex setting. Lin et al. (2018a) study empirically hierarchical variants of local SGD.

Local SGD with averaging in every step, i.e. $H = 1$, is identical to mini-batch SGD. Dekel et al. (2012) show that batch sizes $b = T^{\delta}$, for $\delta \in (0, \frac{1}{2})$ are asymptotically optimal for mini-batch SGD, however they also note that this asymptotic bound might be crude for practical purposes. Similar considerations might also apply to the asymptotic upper bounds on the communication frequency $H$ derived here. Local SGD with averaging only at the end, i.e. $H = T$, is identical to one-shot SGD. Jain et al. (2018) give concise speedup results in terms of bias and variance for one-shot SGD with constant stepsizes for the optimization of quadratic least squares problems. In contrast, our upper bounds become loose when $H \to T$ and our results do not cover one-shot SGD.

Recently, Woodworth et al. (2018) provided a lower bound for parallel stochastic optimization (in the convex setting, and not for strongly convex functions as considered here). The bound is not known to be tight for local SGD.

## 1.3 OUTLINE

We formally introduce local SGD in Section 2 and sketch the convergence proof in Section 3. In Section 4 show numerical results to illustrate the result. We analyze asynchronous local SGD in Section 5. The proof of the technical results, further discussion about the experimental setup and implementation guidelines are deferred to the appendix.

---

**Algorithm 1** LOCAL SGD

---

1: Initialize variables $\mathbf{x}_0^k = \mathbf{x}_0$ for workers $k \in [K]$
2: **for** $t$ in $0 \dots T-1$ **do**
3:     **parallel for** $k \in [K]$ **do**
4:         Sample $i_t^k$ uniformly in $[n]$
5:         **if** $t+1 \in \mathcal{I}_T$ **then**
6:             $\mathbf{x}_{t+1}^k \leftarrow \frac{1}{K} \sum_{k=1}^K \big(\mathbf{x}_t^k - \eta_t \nabla f_{i_t^k}(\mathbf{x}_t^k)\big)$         ▷ global synchronization
7:         **else**
8:             $\mathbf{x}_{t+1}^k \leftarrow \mathbf{x}_t^k - \eta_t \nabla f_{i_t^k}(\mathbf{x}_t^k)$         ▷ local update
9:         **end if**
10:     **end parallel for**
11: **end for**

---

## 2   LOCAL SGD

The algorithm local SGD (depicted in Algorithm 1) generates in parallel $K$ sequences $\{\mathbf{x}_t^k\}_{t=0}^T$ of iterates, $k \in [K]$. Here $K$ denotes the level of parallelization, i.e. the number of distinct parallel sequences and $T$ the number of steps (i.e. the total number of stochastic gradient evaluations is $TK$). Let $\mathcal{I}_T \subseteq [T]$ with $T \in \mathcal{I}_T$ denote a set of *synchronization indices*. Then local SGD evolves the sequences $\{\mathbf{x}_t^k\}_{t=0}^T$ in the following way:

$$\mathbf{x}_{t+1}^k := \begin{cases} \mathbf{x}_t^k - \eta_t \nabla f_{i_t^k}(\mathbf{x}_t^k), & \text{if } t+1 \notin \mathcal{I}_T \\ \frac{1}{K} \sum_{k=1}^K \big(\mathbf{x}_t^k - \eta_t \nabla f_{i_t^k}(\mathbf{x}_t^k)\big) & \text{if } t+1 \in \mathcal{I}_T \end{cases} \tag{4}$$

where indices $i_t^k \sim_{\text{u.a.r.}} [n]$ and $\{\eta_t\}_{t \geq 0}$ denotes a sequence of stepsizes. If $\mathcal{I}_T = [T]$ then the synchronization of the sequences is performed every iteration. In this case, (4) amounts to parallel or mini-batch SGD with mini-batch size $K$.[4] On the other extreme, if $\mathcal{I}_T = \{T\}$, the synchronization only happens at the end, which is known as *one-shot averaging*.

In order to measure the longest interval between subsequent synchronization steps, we introduce the *gap* of a set of integers.

**Definition 2.1** (gap). *The* gap *of a set* $\mathcal{P} := \{p_0, \dots, p_t\}$ *of* $t+1$ *integers,* $p_i \leq p_{i+1}$ *for* $i = 0, \dots, t-1$, *is defined as* $\text{gap}(\mathcal{P}) := \max_{i=1,\dots,t}(p_i - p_{i-1})$.

### 2.1   VARIANCE REDUCTION IN LOCAL SGD

Before jumping to the convergence result, we first discuss an important observation.

**Parallel (mini-batch) SGD.** For carefully chosen stepsizes $\eta_t$, SGD converges at rate $\mathcal{O}\big(\frac{\sigma^2}{T}\big)$[5] on strongly convex and smooth functions $f$, where $\sigma^2 \geq \mathbb{E}\|\nabla f_{i_t^k}(\mathbf{x}_t^k) - \nabla f(\mathbf{x}_t^k)\|^2$ for $t > 0, k \in [K]$ is an upper bound on the variance, see for instance (Zhao & Zhang, 2015). By averaging $K$ stochastic gradients—such as in parallel SGD—the variance decreases by a factor of $K$, and we conclude that parallel SGD converges at a rate $\mathcal{O}\big(\frac{\sigma^2}{TK}\big)$, i.e. achieves a linear speedup.

**Towards local SGD.** For local SGD such a simple argument is elusive. For instance, just capitalizing the convexity of the objective function $f$ is not enough: this will show that the averaged iterate of $K$ independent SGD sequences converges at rate $\mathcal{O}\big(\frac{\sigma^2}{T}\big)$, i.e. no speedup can be shown in this way.

This indicates that one has to show that local SGD decreases the variance $\sigma^2$ instead, similar as in parallel SGD. Suppose the different sequences $\mathbf{x}_t^k$ evolve *close* to each other. Then it is reasonable to assume that averaging the stochastic gradients $\nabla f_{i_t^k}(\mathbf{x}_t^k)$ for all $k \in [K]$ can still yield a reduction in the variance by a factor of $K$—similar as in parallel SGD. Indeed, we will make this statement precise in the proof below.

---

[4]For the ease of presentation, we assume here that each worker in local SGD only processes a mini-batch of size $b = 1$. This can be done without loss of generality, as we discuss later in Remark 2.4.

[5]For the ease of presentation, we here assume that the bias term is negligible compared to the variance term.

## 2.2 Convergence Result and Discussion

**Theorem 2.2.** *Let $f$ be $L$-smooth and $\mu$-strongly convex, $\mathbb{E}_i \left\| \nabla f_i(\mathbf{x}_t^k) - \nabla f(\mathbf{x}_t^k) \right\|^2 \leq \sigma^2$, $\mathbb{E}_i \left\| \nabla f_i(\mathbf{x}_t^k) \right\|^2 \leq G^2$, for $t = 0, \dots, T-1$, where $\{\mathbf{x}_t^k\}_{t=0}^T$ for $k \in [K]$ are generated according to (4) with $\mathrm{gap}(\mathcal{I}_T) \leq H$ and for stepsizes $\eta_t = \frac{4}{\mu(a+t)}$ with shift parameter $a > \max\{16\kappa, H\}$, for $\kappa = \frac{L}{\mu}$. Then*

$$\mathbb{E} f(\hat{\mathbf{x}}_T) - f^\star \leq \frac{\mu a^3}{2 S_T} \left\| \mathbf{x}_0 - \mathbf{x}^\star \right\|^2 + \frac{4T(T+2a)}{\mu K S_T} \sigma^2 + \frac{256 T}{\mu^2 S_T} G^2 H^2 L \,, \tag{5}$$

*where $\hat{\mathbf{x}}_T = \frac{1}{K S_T} \sum_{k=1}^K \sum_{t=0}^{T-1} w_t \mathbf{x}_t^k$, for $w_t = (a+t)^2$ and $S_T = \sum_{t=0}^{T-1} w_t \geq \frac{1}{3} T^3$.*

We were not especially careful to optimize the constants (and the lower order terms) in (5), so we now state the asymptotic result.

**Corollary 2.3.** *Let $\hat{\mathbf{x}}_T$ be as defined as in Theorem 2.2, for parameter $a = \max\{16\kappa, H\}$. Then*

$$\mathbb{E} f(\hat{\mathbf{x}}_T) - f^\star = O\left( \frac{1}{\mu K T} + \frac{\kappa + H}{\mu K T^2} \right) \sigma^2 + O\left( \frac{\kappa H^2}{\mu T^2} + \frac{\kappa^3 + H^3}{\mu T^3} \right) G^2 \,. \tag{6}$$

For the last estimate we used $\mathbb{E} \mu \left\| \mathbf{x}_0 - \mathbf{x}^\star \right\| \leq 2G$ for $\mu$-strongly convex $f$, as derived in (Rakhlin et al., 2012, Lemma 2).

**Remark 2.4** (Mini-batch local SGD). *So far, we assumed that each worker only computes a single stochastic gradient. In mini-batch local SGD, each worker computes a mini-batch of size $b$ in each iteration. This reduces the variance by a factor of $b$, and thus Theorem (2.2) gives the convergence rate of mini-batch local SGD when $\sigma^2$ is replaced by $\frac{\sigma^2}{b}$.*

We now state some consequences of equation (6). For the ease of the exposition we omit the dependency on $L$, $\mu$, $\sigma^2$ and $G^2$ below, but depict the dependency on the local mini-batch size $b$.

**Convergence rate.** For $T$ large enough and assuming $\sigma > 0$, the very first term is dominating in (6) and local SGD converges at rate $O(1/(KTb))$. That is, local SGD achieves a linear speedup in both, the number of workers $K$ and the mini-batch size $b$.

**Global synchronization steps.** It needs to hold $H = O(\sqrt{T/(Kb)})$ to get the linear speedup. This yields a reduction of the number of communication rounds by a factor $O(\sqrt{T/(Kb)})$ compared to parallel mini-batch SGD without hurting the convergence rate.

**Extreme Cases.** We have not optimized the result for extreme settings of $H$, $K$, $L$ or $\sigma$. For instance, we do not recover convergence for the one-shot averaging, i.e. the setting $H = T$ (though convergence for $H = o(T)$, but at a lower rate).

**Unknown Time Horizon/Adaptive Communication Frequency** Zhang et al. (2016) empirically observe that more frequent communication at the beginning of the optimization can help to get faster time-to-accuracy (see also Lin et al. (2018a)). Indeed, when the number of total iterations $T$ is not known beforehand (as it e.g. depends on the target accuracy, cf. (6) and also Section 4 below), then increasing the communication frequency seems to be a good strategy to keep the communication low, why still respecting the constraint $H = O(\sqrt{T/(Kb)})$ for all $T$.

## 3 Proof Outline

We now give the outline of the proof. The proofs of the lemmas are given in Appendix A.

**Perturbed iterate analysis.** Inspired by the perturbed iterate framework of (Mania et al., 2017) we first define a virtual sequence $\{\bar{\mathbf{x}}_t\}_{t \geq 0}$ in the following way:

$$\bar{\mathbf{x}}_0 = \mathbf{x}_0 \,, \qquad\qquad \bar{\mathbf{x}}_t = \frac{1}{K} \sum_{k=1}^K \mathbf{x}_t^k \,, \tag{7}$$

where the sequences $\{\mathbf{x}_t^k\}_{t \geq 0}$ for $k \in [K]$ are the same as in (4). Notice that this sequence never has to be computed explicitly, it is just a tool that we use in the analysis. Further notice that $\bar{\mathbf{x}}_t = \mathbf{x}_t^k$ for

$k \in [K]$ whenever $t \in \mathcal{I}_T$. Especially, when $\mathcal{I}_T = [T]$, then $\bar{\mathbf{x}}_t \equiv \mathbf{x}_t^k$ for every $k \in [K], t \in [T]$. It will be useful to define

$$\mathbf{g}_t := \frac{1}{K} \sum_{k=1}^{K} \nabla f_{i_t^k}(\mathbf{x}_t^k), \qquad\qquad \bar{\mathbf{g}}_t := \frac{1}{K} \sum_{k=1}^{K} \nabla f(\mathbf{x}_t^k). \qquad (8)$$

Observe $\bar{\mathbf{x}}_{t+1} = \bar{\mathbf{x}}_t - \eta_t \mathbf{g}_t$ and $\mathbb{E}\, \mathbf{g}_t = \bar{\mathbf{g}}_t$.

Now the proof proceeds as follows: we show (i) that the virtual sequence $\{\bar{\mathbf{x}}_t\}_{t \geq 0}$ almost behaves like mini-batch SGD with batch size $K$ (Lemma 3.1 and 3.2), and (ii) the true iterates $\{\mathbf{x}_t^k\}_{t \geq 0, k \in [K]}$ do not deviate much from the virtual sequence (Lemma 3.3). These are the main ingredients in the proof. To obtain the rate we exploit a technical lemma from (Stich et al., 2018).

**Lemma 3.1.** *Let $\{\mathbf{x}_t\}_{t \geq 0}$ and $\{\bar{\mathbf{x}}_t\}_{t \geq 0}$ for $k \in [K]$ be defined as in* (4) *and* (7) *and let $f$ be L-smooth and $\mu$-strongly convex and $\eta_t \leq \frac{1}{4L}$. Then*

$$\mathbb{E} \|\bar{\mathbf{x}}_{t+1} - \mathbf{x}^\star\|^2 \leq (1 - \mu \eta_t) \mathbb{E} \|\bar{\mathbf{x}}_t - \mathbf{x}^\star\|^2 + \eta_t^2 \mathbb{E} \|\mathbf{g}_t - \bar{\mathbf{g}}_t\|^2$$

$$- \frac{1}{2}\eta_t \mathbb{E}(f(\bar{\mathbf{x}}_t) - f^\star) + 2\eta_t \frac{L}{K} \sum_{k=1}^{K} \mathbb{E} \|\bar{\mathbf{x}}_t - \mathbf{x}_t^k\|^2. \qquad (9)$$

**Bounding the variance.** From equation (9) it becomes clear that we should derive an upper bound on $\mathbb{E} \|\mathbf{g}_t - \bar{\mathbf{g}}_t\|^2$. We will relate this to the variance $\sigma^2$.

**Lemma 3.2.** *Let $\sigma^2 \geq \mathbb{E}_i \|\nabla f_i(\mathbf{x}_t^k) - \nabla f(\mathbf{x}_t^k)\|^2$ for $k \in [K]$, $t \in [T]$. Then $\mathbb{E} \|\mathbf{g}_t - \bar{\mathbf{g}}_t\|^2 \leq \frac{\sigma^2}{K}$.*

**Bounding the deviation.** Further, we need to bound $\frac{1}{K} \sum_{k=1}^{K} \mathbb{E} \|\bar{\mathbf{x}}_t - \mathbf{x}_t^k\|^2$. For this we impose a condition on $\mathcal{I}_T$ and an additional condition on the stepsize $\eta_t$.

**Lemma 3.3.** *If $\mathrm{gap}(\mathcal{I}_T) \leq H$ and sequence of decreasing positive stepsizes $\{\eta_t\}_{t \geq 0}$ satisfying $\eta_t \leq 2\eta_{t+H}$ for all $t \geq 0$, then*

$$\frac{1}{K} \sum_{k=1}^{K} \mathbb{E} \|\bar{\mathbf{x}}_t - \mathbf{x}_t^k\|^2 \leq 4\eta_t^2 G^2 H^2, \qquad (10)$$

*where $G^2$ is a constant such that $\mathbb{E}_i \|\nabla f_i(\mathbf{x}_t^k)\|^2 \leq G^2$ for $k \in [K], t \in [T]$.*

**Optimal Averaging.** Similar as in (Lacoste-Julien et al., 2012; Shamir & Zhang, 2013; Rakhlin et al., 2012) we define a suitable averaging scheme for the iterates $\{\bar{\mathbf{x}}_t\}_{t \geq 0}$ to get the optimal convergence rate. In contrast to (Lacoste-Julien et al., 2012) that use linearly increasing weights, we use quadratically increasing weights, as for instance (Shamir & Zhang, 2013; Stich et al., 2018).

**Lemma 3.4** ((Stich et al., 2018)). *Let $\{a_t\}_{t \geq 0}$, $a_t \geq 0$, $\{e_t\}_{t \geq 0}$, $e_t \geq 0$ be sequences satisfying*

$$a_{t+1} \leq (1 - \mu \eta_t) a_t - \eta_t e_t A + \eta_t^2 B + \eta_t^3 C, \qquad (11)$$

*for $\eta_t = \frac{4}{\mu(a+t)}$ and constants $A > 0$, $B, C \geq 0$, $\mu > 0$, $a > 1$. Then*

$$\frac{A}{S_T} \sum_{t=0}^{T-1} w_t e_t \leq \frac{\mu a^3}{4 S_T} a_0 + \frac{2T(T + 2a)}{\mu S_T} B + \frac{16T}{\mu^2 S_T} C, \qquad (12)$$

*for $w_t = (a + t)^2$ and $S_T := \sum_{t=0}^{T-1} w_t = \frac{T}{6}\left(2T^2 + 6aT - 3T + 6a^2 - 6a + 1\right) \geq \frac{1}{3}T^3$.*

*Proof.* This is a reformulation of Lemma 3.3 in (Stich et al., 2018). $\qquad\square$

**Proof of Theorem 2.2.** By convexity of $f$ we have $\mathbb{E}\, f(\hat{\mathbf{x}}_T) - f^\star \leq \frac{1}{S_T} \sum_{t=0}^{T-1} w_t \mathbb{E}\big(f(\bar{\mathbf{x}}_t) - f^\star\big)$. The proof of the theorem thus follows immediately from the four lemmas that we have presented, i.e. by Lemma 3.4 with $e_t := \mathbb{E}(f(\bar{\mathbf{x}}_t) - f^\star)$ and constants $A = \frac{1}{2}$, (Lemma 3.1), $B = \frac{\sigma^2}{K}$, (Lemma 3.2) and $C = 8G^2 H^2 L$, (Lemma 3.3). Observe that the stepsizes $\eta_t = \frac{4}{\mu(a+t)}$ satisfy both the conditions of Lemma 3.1 ($\eta_0 = \frac{4}{\mu a} \leq \frac{1}{4L}$, as $a \geq 16\kappa$) and of Lemma 3.3 $\left(\frac{\eta_t}{\eta_{t+H}} = \frac{a+t+H}{a+t} \leq 2\right.$, as $a \geq H\left.\right)$. $\square$

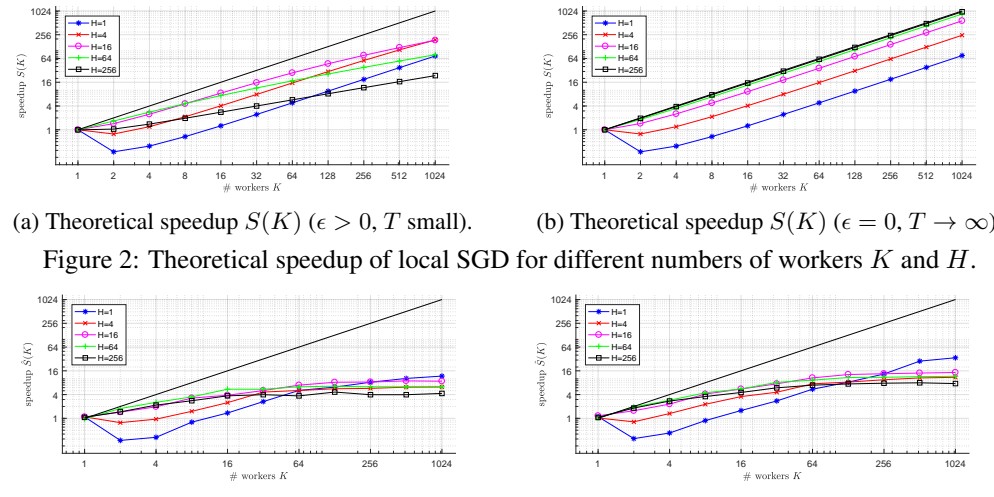

(a) Theoretical speedup $S(K)$ ($\epsilon > 0, T$ small).  (b) Theoretical speedup $S(K)$ ($\epsilon = 0, T \to \infty$).

Figure 2: Theoretical speedup of local SGD for different numbers of workers $K$ and $H$.

(a) Measured speedup, $\epsilon = 0.005$.  (b) Measured speedup, $\epsilon = 0.0001$.

Figure 3: Measured speedup of local SGD with mini-batch $b = 4$ for different numbers of workers $K$ and parameters $H$.

## 4 NUMERICAL ILLUSTRATION

In this section we show some numerical experiments to illustrate the results of Theorem 2.2.

**Speedup.** When Algorithm 1 is implemented in a distributed setting, there are two components that determine the wall-clock time: (i) the total number of gradient computations, $TK$, and (ii) the total time spend for communication. In each communication round $2(K - 1)$ vectors need to be exchanged, and there will be $T/H$ communication rounds. Typically, the communication is more expensive than a single gradient computation. We will denote this ratio by a factor $\rho \geq 1$ (in practice, $\rho$ can be 10–100, or even larger on slow networks). The parameter $T$ depends on the desired accuracy $\epsilon > 0$, and according to (6) we roughly have $T(\epsilon, H, K) \approx \frac{1}{K\epsilon}\left(\frac{1}{2} + \frac{1}{2}\sqrt{1 + \epsilon(1 + H + H^2K)}\right)$. Thus, the theoretical speedup $S(K)$ of local SGD on $K$ machines compared to SGD on one machine ($H = 1, K = 1$) is

$$S(K) = \frac{K}{\left(\frac{1}{2} + \frac{1}{2}\sqrt{1 + \epsilon(1 + H + H^2K)}\right)\left(1 + 2\rho\frac{(K-1)}{H}\right)}. \tag{13}$$

**Theoretical.** Examining (13), we see that (i) increasing $H$ can reduce negative scaling effects due to parallelization (second bracket in the denominator of (13)), and (ii) local SGD only shows linear scaling for $\epsilon \ll 1$ (i.e. $T$ large enough, in agreement with the theory). In Figure 2 we depict $S(K)$, once for $\epsilon = 0$ in Figure 2b, and for positive $\epsilon > 0$ in Figure 2a under the assumption $\rho = 25$. We see that for $\epsilon = 0$ the largest values of $H$ give the best speedup, however, when only a few epochs need to be performed, then the optimal values of $H$ change with the number of workers $K$. We also see that for a small number of workers $H = 1$ is never optimal. If $T$ is unknown, then these observations seem to indicate that the technique from (Zhang et al., 2016), i.e. adaptively increasing $H$ over time seems to be a good strategy to get the best choice of $H$ when the time horizon is unknown.

**Experimental.** We examine the practical speedup on a logistic regression problem, $f(\mathbf{x}) = \frac{1}{n}\sum_{i=1}^{n}\log(1 + \exp(-b_i\mathbf{a}_i^\top\mathbf{x})) + \frac{\lambda}{2}\|\mathbf{x}\|^2$, where $\mathbf{a}_i \in \mathbb{R}^d$ and $b_i \in \{-1, +1\}$ are the data samples. The regularization parameter is set to $\lambda = 1/n$. We consider the w8a dataset (Platt, 1999) ($d = 300, n = 49749$). We initialize all runs with $\mathbf{x}_0 = \mathbf{0}_d$ and measure the number of iterations to reach the target accuracy $\epsilon$. We consider the target accuracy reached, when either the last iterate, the uniform average, the average with linear weights, or the average with quadratic weights (such as in Theorem 2.2) reaches the target accuracy. By extensive grid search we determine for each configuration $(H, K, B)$ the best stepsize from the set $\{\min(32, \frac{cn}{t+1}), 32c\}$, where $c$ can take the values $c = 2^i$ for $i \in \mathbb{Z}$. For more details on the experimental setup refer Section D in the appendix. We depict the results in Figure 3, again under the assumption $\rho = 25$.

---

**Algorithm 2** ASYNCHRONOUS LOCAL SGD (SCHEMATIC)

---

1: Initialize variables $\mathbf{x}_0^k = \mathbf{x}_0$, $r^k = 0$ for $k \in [K]$, aggregate $\bar{\bar{\mathbf{x}}} = \mathbf{x}_0$.
2: **parallel for** $k \in [K]$ **do**
3:   **for** $t$ in $0 \ldots T - 1$ **do**
4:     Sample $i_t^k$ uniformly in $[n]$
5:     $\mathbf{x}_{t+1}^k \leftarrow \mathbf{x}_t^k - \eta_t \nabla f_{i_t^k}(\mathbf{x}_t^k)$                    ▷ local update
6:     **if** $t + 1 \in \mathcal{I}_T^k$ **then**
7:       $\bar{\bar{\mathbf{x}}} \leftarrow \mathrm{add}(\bar{\bar{\mathbf{x}}}, \frac{1}{K}(\mathbf{x}_{t+1}^k - \mathbf{x}_{r^k}^k))$             ▷ atomic aggregation of the updates
8:       $\mathbf{x}_{t+1}^k \leftarrow \mathrm{read}(\bar{\bar{\mathbf{x}}})$;
9:       $r^k \leftarrow t + 1$                            ▷ iteration/time of last read
10:     **end if**
11:   **end for**
12: **end parallel for**

---

**Conclusion.**   The restriction on $H$ imposed by theory is not severe for $T \to \infty$. Thus, for training that either requires many passes over the data or that is performed only on a small cluster, large values of $H$ are advisable. However, for smaller $T$ (few passes over the data), the $O(1/\sqrt{K})$ dependency shows significantly in the experiment. This has to be taken into account when deploying the algorithm on a massively parallel system, for instance through the technique mentioned in (Zhang et al., 2016).

## 5   ASYNCHRONOUS LOCAL SGD

In this section we present asynchronous local SGD that does not require that the local sequences are synchronized. This does not only reduce communication bottlenecks, but by using load-balancing techniques the algorithm can optimally be tuned to heterogeneous settings (slower workers do less computation between synchronization, and faster workers do more). We will discuss this in more detail in Section C.

Asynchronous local SGD generates in parallel $K$ sequences $\{\mathbf{x}_t^k\}_{t=0}^T$ of iterates, $k \in [K]$. Similar as in Section 2 we introduce sets of synchronization indices, $\mathcal{I}_t^k \subseteq [T]$ with $T \in \mathcal{I}_T^k$ for $k \in [K]$. Note that the sets do not have to be equal for different workers. Each worker $k$ evolves locally a sequence $\mathbf{x}_t^k$ in the following way:

$$\mathbf{x}_{t+1}^k = \begin{cases} \mathbf{x}_t^k - \gamma_t \nabla f_{i_t^k}(\mathbf{x}_t^k) & \text{if } t + 1 \notin \mathcal{I}_T^k \\ \bar{\bar{\mathbf{x}}}_{t+1}^k & \text{if } t + 1 \in \mathcal{I}_T^k \end{cases} \tag{14}$$

where $\bar{\bar{\mathbf{x}}}_{t+1}^k$ denotes the state of the aggregated variable at the time when worker $k$ reads the aggregated variable. To be precise, we use the notation

$$\bar{\bar{\mathbf{x}}}_t^k = \mathbf{x}_0 - \frac{1}{K} \sum_{h=1}^K \sum_{j=0}^{t-1} \mathbb{1}_{j \in \mathcal{W}_t^{k,h}} (\gamma_j \nabla f_{i_j^k}(\mathbf{x}_j^k)), \tag{15}$$

where $\mathcal{W}_t^{k,h} \subseteq [T]$ denotes all updates that have been written at the time the read takes place. The sets $\mathcal{W}_t^{k,h}$ are indexed by iteration $t$, worker $k$ that initiates the read and $h \in [K]$. Thus $\mathcal{W}_t^{k,h}$ denotes all updates of the local sequence $\{\mathbf{x}_t^h\}_{t \geq 0}$, that have been reported back to the server at the time worker $k$ reads (in iteration $t$). This notation is necessary, as we don't necessarily have $\mathcal{W}_t^{k,h} = \mathcal{W}_t^{k',h}$ for $k \neq k'$. We have $\mathcal{W}_t^{k,h} \subseteq \mathcal{W}_{t'}^{k,h}$ for $t' \geq t$, as updates are not overwritten. When we cast synchronized local SGD in this notation, then it holds $\mathcal{W}_t^{k,h} = \mathcal{W}_t^{k',h'}$ for all $k, h, k', h'$, as all the writes and reads are synchronized.

**Theorem 5.1.** *Let $f$, $\sigma$, $G$ and $\kappa$ be as in Theorem 5.1 and sequences $\{\mathbf{x}_t^k\}_{t=0}^T$ for $k \in [K]$ generated according to (14) with $\mathrm{gap}(\mathcal{I}_T^k) \leq H$ for $k \in K$ and for stepsizes $\eta_t = \frac{4}{\mu(a+t)}$ with shift parameter $a > \max\{16\kappa, H + \tau\}$ for delay $\tau > 0$. If $\mathcal{W}_t^{k,h} \supseteq [t - \tau]$ for all $k, h \in [K]$, $t \in [T]$, then*

$$\mathbb{E} f(\hat{\mathbf{x}}_T) - f^\star \leq \frac{\mu a^3}{2S_T} \|\mathbf{x}_0 - \mathbf{x}^\star\|^2 + \frac{4T(T + 2a)}{\mu K S_T} \sigma^2 + \frac{768T}{\mu^2 S_T} G^2 (H + \sigma)^2 L, \tag{16}$$

*where $\hat{\mathbf{x}}_T = \frac{1}{K S_T} \sum_{k=1}^K \sum_{t=0}^{T-1} w_t \mathbf{x}_t^k$, for $w_t = (a + t)^2$ and $S_T = \sum_{t=0}^{T-1} w_t \geq \frac{1}{3} T^3$.*

Hence, for $T$ large enough and $(H + \tau) = O(\sqrt{T/K})$, asynchronous local SGD converges with rate $O\left(\frac{G^2}{KT}\right)$, the same rate as synchronous local SGD.

## 6  CONCLUSION

We prove convergence of synchronous and asynchronous local SGD and are the first to show that local SGD (for nontrivial values of $H$) attains theoretically linear speedup on strongly convex functions when parallelized among $K$ workers. We show that local SGD saves up to a factor of $O(T^{1/2})$ in global communication rounds compared to mini-batch SGD, while still converging at the same rate in terms of total stochastic gradient computations.

Deriving more concise convergence rates for local SGD could be an interesting future direction that could deepen our understanding of the scheme. For instance one could aim for a more fine grained analysis in terms of bias and variance terms (similar as e.g. in Dekel et al. (2012); Jain et al. (2018)), relaxing the assumptions (here we relied on the bounded gradient assumption), or investigating the data dependence (e.g. by considering data-depentent measures like e.g. gradient diversity Yin et al. (2018)). There are also no apparent reasons that would limit the extension of the theory to non-convex objective functions; Lemma 3.3 does neither use the smoothness nor the strong convexity assumption, so this can be applied in the non-convex setting as well. We feel that the positive results shown here can motivate and spark further research on non-convex problems. Indeed, very recent work (Zhou & Cong, 2018; Yu et al., 2018) analyzes local SGD for non-convex optimization problems and shows convergence of SGD to a stationary point, though the restrictions on $H$ are stronger than here.

## ACKNOWLEDGMENTS

The author thanks Jean-Baptiste Cordonnier, Tao Lin and Kumar Kshitij Patel for spotting various typos in the first versions of this manuscript, as well as Martin Jaggi for his support.

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

## A  MISSING PROOFS FOR SYNCHRONIZED LOCAL SGD

In this section we provide the proofs for the three lemmas that were introduced in Section 3.

**Proof of Lemma 3.1.** Using the update equation (7) we have

$$\left\|\bar{\mathbf{x}}_{t+1} - \mathbf{x}^\star\right\|^2 = \left\|\bar{\mathbf{x}}_t - \eta_t \mathbf{g}_t - \mathbf{x}^\star\right\|^2 = \left\|\bar{\mathbf{x}}_t - \eta_t \mathbf{g}_t - \mathbf{x}^\star - \eta_t \bar{\mathbf{g}}_t + \eta_t \bar{\mathbf{g}}_t\right\|^2 \tag{17}$$

$$= \left\|\bar{\mathbf{x}}_t - \mathbf{x}^\star - \eta_t \bar{\mathbf{g}}_t\right\|^2 + \eta_t^2 \left\|\mathbf{g}_t - \bar{\mathbf{g}}_t\right\|^2 + 2\eta_t \left\langle \bar{\mathbf{x}}_t - \mathbf{x}^\star - \eta_t \bar{\mathbf{g}}_t, \bar{\mathbf{g}}_t - \mathbf{g}_t \right\rangle . \tag{18}$$

Observe that

$$\left\|\bar{\mathbf{x}}_t - \mathbf{x}^\star - \eta_t \bar{\mathbf{g}}_t\right\|^2 = \left\|\bar{\mathbf{x}}_t - \mathbf{x}^\star\right\|^2 + \eta_t^2 \left\|\bar{\mathbf{g}}_t\right\|^2 - 2\eta_t \left\langle \bar{\mathbf{x}}_t - \mathbf{x}^\star, \bar{\mathbf{g}}_t \right\rangle \tag{19}$$

$$= \left\|\bar{\mathbf{x}}_t - \mathbf{x}^\star\right\|^2 + \eta_t^2 \left\|\bar{\mathbf{g}}_t\right\|^2 - 2\eta_t \frac{1}{K} \sum_{k=1}^K \left\langle \bar{\mathbf{x}}_t - \mathbf{x}^\star, \nabla f(\mathbf{x}_t^k) \right\rangle \tag{20}$$

$$\leq \left\|\bar{\mathbf{x}}_t - \mathbf{x}^\star\right\|^2 + \eta_t^2 \frac{1}{K} \sum_{k=1}^K \left\|\nabla f(\mathbf{x}_t^k)\right\|^2$$

$$- 2\eta_t \frac{1}{K} \sum_{k=1}^K \left\langle \bar{\mathbf{x}}_t - \mathbf{x}_k^t + \mathbf{x}_k^t - \mathbf{x}^\star, \nabla f(\mathbf{x}_t^k) \right\rangle \tag{21}$$

$$= \left\|\bar{\mathbf{x}}_t - \mathbf{x}^\star\right\|^2 + \eta_t^2 \frac{1}{K} \sum_{k=1}^K \left\|\nabla f(\mathbf{x}_t^k) - \nabla f(\mathbf{x}^\star)\right\|^2$$

$$- 2\eta_t \frac{1}{K} \sum_{k=1}^K \left\langle \mathbf{x}_k^t - \mathbf{x}^\star, \nabla f(\mathbf{x}_t^k) \right\rangle - 2\eta_t \frac{1}{K} \sum_{k=1}^K \left\langle \bar{\mathbf{x}}_t - \mathbf{x}_k^t, \nabla f(\mathbf{x}_t^k) \right\rangle , \tag{22}$$

where we used the inequality $\left\|\sum_{i=1}^K \mathbf{a}_i\right\|^2 \leq K \sum_{i=1}^K \left\|\mathbf{a}_i\right\|^2$ in (21). By $L$-smoothness,

$$\left\|\nabla f(\mathbf{x}_t^k) - \nabla f(\mathbf{x}^\star)\right\|^2 \leq 2L(f(\mathbf{x}_t^k) - f^\star) , \tag{23}$$

and by $\mu$-strong convexity

$$- \left\langle \mathbf{x}_t^k - \mathbf{x}^\star, \nabla f(\mathbf{x}_t^k) \right\rangle \leq -(f(\mathbf{x}_t^k) - f^\star) - \frac{\mu}{2} \left\|\mathbf{x}_t^k - \mathbf{x}^\star\right\|^2 . \tag{24}$$

To estimate the last term in (22) we use $2 \left\langle \mathbf{a}, \mathbf{b} \right\rangle \leq \gamma \left\|\mathbf{a}\right\|^2 + \gamma^{-1} \left\|\mathbf{b}\right\|^2$, for $\gamma > 0$. This gives

$$-2 \left\langle \bar{\mathbf{x}}_t - \mathbf{x}_k^t, \nabla f(\mathbf{x}_t^k) \right\rangle \leq 2L \left\|\bar{\mathbf{x}}_t - \mathbf{x}_k^t\right\|^2 + \frac{1}{2L} \left\|\nabla f(\mathbf{x}_t^k)\right\|^2 \tag{25}$$

$$= 2L \left\|\bar{\mathbf{x}}_t - \mathbf{x}_k^t\right\|^2 + \frac{1}{2L} \left\|\nabla f(\mathbf{x}_t^k) - \nabla f(\mathbf{x}^\star)\right\|^2 \tag{26}$$

$$\leq 2L \left\|\bar{\mathbf{x}}_t - \mathbf{x}_k^t\right\|^2 + (f(\mathbf{x}_t^k) - f^\star) , \tag{27}$$

where we have again used (23) in the last inequality. By applying these three estimates to (22) we get

$$\left\|\bar{\mathbf{x}}_t - \mathbf{x}^\star - \eta_t \bar{\mathbf{g}}_t\right\|^2 \leq \left\|\bar{\mathbf{x}}_t - \mathbf{x}^\star\right\|^2 + 2\eta_t \frac{L}{K} \sum_{k=1}^K \left\|\bar{\mathbf{x}}_t - \mathbf{x}_k^t\right\|^2$$

$$+ 2\eta_t \frac{1}{K} \sum_{k=1}^K \left( \left( \eta_t L - \frac{1}{2} \right) (f(\mathbf{x}_t^k) - f^\star) - \frac{\mu}{2} \left\|\mathbf{x}_t^k - \mathbf{x}^\star\right\|^2 \right) . \tag{28}$$

For $\eta_t \leq \frac{1}{4L}$ it holds $\left( \eta_t L - \frac{1}{2} \right) \leq -\frac{1}{4}$. By convexity of $a \left( f(\mathbf{x}) - f^\star \right) + b \left\|\mathbf{x} - \mathbf{x}^\star\right\|^2$ for $a, b \geq 0$:

$$-\frac{1}{K} \sum_{k=1}^K \left( a(f(\mathbf{x}_t^k) - f^\star) + b \left\|\mathbf{x}_t^k - \mathbf{x}^\star\right\|^2 \right) \leq - \left( a(f(\bar{\mathbf{x}}_t) - f^\star) + b \left\|\bar{\mathbf{x}}_t - \mathbf{x}^\star\right\|^2 \right) , \tag{29}$$

hence we can continue in (28) and obtain

$$\left\| \bar{\mathbf{x}}_t - \mathbf{x}^\star - \eta_t \bar{\mathbf{g}}_t \right\|^2 \leq (1 - \mu \eta_t) \left\| \bar{\mathbf{x}}_t - \mathbf{x}^\star \right\|^2 - \frac{1}{2} \eta_t (f(\bar{\mathbf{x}}_t) - f^\star) + 2\eta_t \frac{L}{K} \sum_{k=1}^{K} \left\| \bar{\mathbf{x}}_t - \mathbf{x}_t^k \right\|^2 . \quad (30)$$

Finally, we can plug (30) back into (18). By taking expectation we get

$$\mathbb{E} \left\| \bar{\mathbf{x}}_{t+1} - \mathbf{x}^\star \right\|^2 \leq (1 - \mu \eta_t) \mathbb{E} \left\| \bar{\mathbf{x}}_t - \mathbf{x}^\star \right\|^2 + \eta_t^2 \, \mathbb{E} \left\| \mathbf{g}_t - \bar{\mathbf{g}}_t \right\|^2$$
$$- \frac{1}{2} \eta_t \, \mathbb{E}(f(\bar{\mathbf{x}}_t) - f^\star) + 2\eta_t \frac{L}{K} \sum_{k=1}^{K} \mathbb{E} \left\| \bar{\mathbf{x}}_t - \mathbf{x}_t^k \right\|^2 . \qquad \square$$

**Proof of Lemma 3.2.** By definition of $\mathbf{g}_t$ and $\bar{\mathbf{g}}_t$ we have

$$\mathbb{E} \left\| \mathbf{g}_t - \bar{\mathbf{g}}_t \right\|^2 = \mathbb{E} \left\| \frac{1}{K} \sum_{k=1}^{K} \left( \nabla f_{i_t^k}(\mathbf{x}_t^k) - \nabla f(\mathbf{x}_t^k) \right) \right\|^2 = \frac{1}{K^2} \sum_{k=1}^{K} \mathbb{E} \left\| \nabla f_{i_t^k}(\mathbf{x}_t^k) - \nabla f(\mathbf{x}_t^k) \right\|^2 \leq \frac{\sigma^2}{K},$$
$$(31)$$

where we used $\mathrm{Var}(\sum_{k=1}^{K} X_k) = \sum_{k=1}^{K} \mathrm{Var}(X_k)$ for independent random variables. $\qquad \square$

**Proof of Lemma 3.3.** As the $\mathrm{gap}(\mathcal{I}_T) \leq H$, there is an index $t_0$, $t - t_0 \leq H$ such that $\bar{\mathbf{x}}_{t_0} = \mathbf{x}_{t_0}^k$ for $k \in [K]$. Observe, using $\mathbb{E} \left\| X - \mathbb{E} X \right\|^2 = \mathbb{E} \left\| X \right\|^2 - \left\| \mathbb{E} X \right\|^2$ and $\left\| \sum_{i=1}^{H} \mathbf{a}_i \right\|^2 \leq H \sum_{i=1}^{H} \left\| \mathbf{a}_i \right\|^2$,

$$\frac{1}{K} \sum_{k=1}^{K} \mathbb{E} \left\| \bar{\mathbf{x}}_t - \mathbf{x}_t^k \right\|^2 = \frac{1}{K} \sum_{k=1}^{K} \mathbb{E} \left\| \mathbf{x}_t^k - \mathbf{x}_{t_0} - (\bar{\mathbf{x}}_t - \mathbf{x}_{t_0}) \right\|^2 \qquad (32)$$

$$\leq \frac{1}{K} \sum_{k=1}^{K} \mathbb{E} \left\| \mathbf{x}_t^k - \mathbf{x}_{t_0} \right\|^2 \qquad (33)$$

$$\leq \frac{1}{K} \sum_{k=1}^{K} H \eta_{t_0}^2 \sum_{h=t_0}^{t-1} \mathbb{E} \left\| \nabla f_{i_h^k}(x_h^k) \right\|^2 \qquad (34)$$

$$\leq \frac{1}{K} \sum_{k=1}^{K} H^2 \eta_{t_0}^2 G^2 , \qquad (35)$$

where we used $\eta_t \leq \eta_{t_0}$ for $t \geq t_0$ and the assumption $\mathbb{E} \| \nabla f_{i_h^k}(\mathbf{x}_h^k) \|^2 \leq G^2$. Finally, the claim follows by the assumption on the stepsizes, $\frac{\eta_{t_0}}{\eta_t} \leq 2$. $\qquad \square$

## B    MISSING PROOF FOR ASYNCHRONOUS LOCAL SGD

In this Section we prove Theorem 5.1. The proof follows closely the proof presented in Section 3. We again introduce the virtual sequence

$$\bar{\mathbf{x}}_t = \mathbf{x}_0 - \frac{1}{K} \sum_{h=1}^{K} \sum_{j=0}^{t-1} \eta_j \nabla f_{i_j^k}(\mathbf{x}_j^k) , \qquad (36)$$

as before. By the property $T \in \mathcal{I}_T^k$ for $k \in K$ we know that all workers will have written their updates when the algorithm terminates. This assumption is not very critical and could be relaxed, but it facilitates the (already quite heavy) notation in the proof.

Observe, that Lemmas 3.1 and 3.2 hold for the virtual sequence $\{\bar{\mathbf{x}}_t\}_{t=0}^{T}$. Hence, all we need is a refined version of Lemma 3.3 that bounds how far the local sequences can deviate from the virtual average.

**Lemma B.1.** *If* $\mathrm{gap}(\mathcal{I}_T^k) \le H$ *and* $\exists \tau > 0$, *s.t.* $\mathcal{W}_t^{k,h} \supseteq [t - \tau]$ *for all* $k, h \in [K]$, $t \in [T]$, *and sequence of decreasing positive stepsizes* $\{\eta_t\}_{t \ge 0}$ *satisfying* $\eta_t \le 2\eta_{t+H+\tau}$ *for all* $t \ge 0$, *then*

$$\frac{1}{K} \sum_{k=1}^{K} \mathbb{E} \left\| \bar{\mathbf{x}}_t - \mathbf{x}_t^k \right\|^2 \le 12 \eta_t^2 G^2 (H + \tau)^2 \,, \tag{37}$$

*where* $G^2$ *is a constant such that* $\mathbb{E}_i \|\nabla f_i(\mathbf{x}_t^k)\|^2 \le G^2$ *for* $k \in [K], t \in [T]$.

Here we use the notation $[s] = \{\}$ for $s < 0$, such that $[t - \tau]$ is also defined for $t < \tau$.

*Proof.* As $\mathrm{gap}(\mathcal{I}_T^k) \le H$ there exists for every $k \in K$ a $t_k$, $t - t_k \le H$, such that $\mathbf{x}_{t_k}^k = \bar{\bar{x}}_{t_k}^k$. Let $t_0 := \min\{t_1, \ldots, t_K\}$ and observe $t_0 \ge t - H$. Let $t_0' = \max\{t_0 - \tau, 0\}$. As $\mathcal{W}_t^{k,h} \supseteq [t - \tau]$ for all $k, h \in [K], t \in [T]$, it holds

$$\bar{\bar{\mathbf{x}}}_{t_k}^k = \bar{\mathbf{x}}_{t_0'} - \frac{1}{K} \sum_{h=1}^{K} \sum_{j=t_0'}^{t_k - 1} \mathbb{1}_{j \in \mathcal{W}_{t_k}^{k,h}} (\eta_j \nabla f_{i_j^k}(\mathbf{x}_j^k)) \,, \tag{38}$$

for each $k \in [K]$. In other words, all updates up to iteration $t_0'$ have been written to the aggregated sequence.

We decompose the error term as

$$\left\| \bar{\mathbf{x}}_t - \mathbf{x}_t^k \right\|^2 \le 3 \left( \left\| \mathbf{x}_t^k - \mathbf{x}_{t_k}^k \right\|^2 + \left\| \mathbf{x}_{t_k}^k - \bar{\mathbf{x}}_{t_0'} \right\|^2 + \left\| \bar{\mathbf{x}}_{t_0'} - \bar{\mathbf{x}}_t \right\|^2 \right) \,. \tag{39}$$

Now, using $\eta_t \ge \eta_{t+1}$, and $t - t_k \le H$, we conclude (as in (35))

$$\left\| \mathbf{x}_t^k - \mathbf{x}_{t_k}^k \right\|^2 \le \eta_{t_k}^2 H^2 G^2 \le \eta_{t_0}^2 H^2 G^2 \,. \tag{40}$$

As $t_k - t_0' \le \tau$,

$$\left\| \mathbf{x}_{t_k}^k - \bar{\mathbf{x}}_{t_0'} \right\|^2 \le \eta_{t_0'}^2 \tau^2 G^2 \,, \tag{41}$$

and similarly, as $t - t_0' \le H + \tau$,

$$\left\| \tilde{\mathbf{x}}_{t_0'} - \tilde{\mathbf{x}}_t \right\|^2 \le \eta_{t_0'}^2 (H + \tau)^2 G^2 \,. \tag{42}$$

Finally, as $\frac{\eta_{t_0'}}{\eta_t} \le 2$, we can conclude

$$\left\| \bar{\mathbf{x}}_t - \mathbf{x}_t^k \right\|^2 \le 12 \eta_t^2 (H + \tau)^2 G^2 \,. \tag{43}$$

and the lemma follows. $\qquad \square$

Now the proof of Theorem 5.1 follows immediately.

*Proof of Theorem 5.1.* As in the proof of Theorem 2.2 we rely on Lemma 3.4 to derive the convergence rate. Again, we have $A = \frac{1}{2}$, $B = \frac{\sigma^2}{K}$, and $C = LG^2(H + \tau)^2$ (Lemma B.1). It is easy to see that the stepsizes satisfy the condition of Lemma B.1, as clearly $\frac{\eta_{t_0'}}{\eta_t} \le \frac{\eta_{t_0'}}{\eta_{t_0' + H + \tau}} = \frac{a + t + H + \tau}{a + t} \le 2$, as $a \ge H + \tau$. $\qquad \square$

# C   COMMENTS ON IMPLEMENTATION ISSUES

## C.1   SYNCHRONOUS LOCAL SGD

In Theorem 5 we do not prove convergence of the sequences $\{\mathbf{x}_t^k\}_{t \ge 0}$ of the iterates, but only convergence of a weighted average of all iterates. In practice, the last iterate might often be sufficient, but we like to remark that the weighted average of the iterates can easily be tracked on the fly with an auxiliary sequence $\{\mathbf{y}_t\}_{t>0}$, $\mathbf{y}_0 = \mathbf{x}_0$, without storing all intermediate iterates, see Table 1 for some examples.

| criteria | weights | formula | recursive update |
|---|---|---|---|
| last iterate | - | $\mathbf{y}_t = \mathbf{x}_t$ | $\mathbf{y}_t = \mathbf{x}_t$ |
| uniform average | $w_t = 1$ | $\mathbf{y}_t = \frac{1}{t+1}\sum_{i=0}^t \mathbf{x}_i$ | $\mathbf{y}_t = \frac{1}{t+1}\mathbf{x}_t + \frac{t}{t+1}\mathbf{y}_{t-1}$ |
| linear weights | $w_t = (t+1)$ | $\mathbf{y}_t = \frac{2}{(1+t)(2+t)}\sum_{i=0}^t (i+1)\mathbf{x}_i$ | $\mathbf{y}_t = \frac{2}{2+t}\mathbf{x}_t + \frac{t}{t+2}\mathbf{y}_{t-1}$ |
| quadratic weights | $w_t = (t+1)^2$ | $\mathbf{y}_t = \frac{6}{(t+1)(t+2)(2t+3)}\sum_{i=0}^t (i+1)^2\mathbf{x}_i$ | $\mathbf{y}_t = \frac{6(t+1)}{(t+2)(2t+3)}\mathbf{x}_t + \frac{t(1+2t)}{6+7t+2t^2}\mathbf{y}_{t-1}$ |

Table 1: Formulas to recursively track weighted averages.

## C.2 ASYNCHRONOUS LOCAL SGD

As for synchronous local SGD, the weighted averages of the iterates (if needed), can be tracked on each worker locally by a recursive formula as explained above.

A more important aspect that we do not have discussed yet, is that Algorithm 2 allows for an easy procedure to balance the load in heterogeneous settings. In our notation, we have always associated the local sequences $\{\mathbf{x}_t^k\}$ with a specific worker $k$. However, the computation of the sequences does not need to be tied to a specific worker. Thus, a fast worker $k$ that has advanced his local sequence too much already, can start computing updates for another sequence $k' \neq k$, if worker $k'$ is lagged behind. This was not possible in the synchronous model, as there all communications had to happen in sync. We demonstrate this principle in Table 2 below for two workers. Note that also the running averages can still be maintained.

| wall clock time | $\rightarrow$ | $\rightarrow$ | $\rightarrow$ | $\rightarrow$ | $\rightarrow$ | $\rightarrow$ |
|---|---|---|---|---|---|---|
| worker 1 | $\mathbf{x}_H^1 \leftarrow U(\bar{\mathbf{x}})$ | $\mathbf{x}_{2H}^1 \leftarrow U(\bar{\mathbf{x}})$ | $\mathbf{x}_{3H}^1 \leftarrow U(\bar{\mathbf{x}})$ | $\mathbf{x}_{2H}^2 \leftarrow U(\bar{\mathbf{x}})$ | $\mathbf{x}_{4H}^2 \leftarrow U(\bar{\mathbf{x}})$ | $\mathbf{x}_{4H}^1 \leftarrow U(\bar{\mathbf{x}})$ $\cdots$ |
| worker 2 | | $\mathbf{x}_H^2 \leftarrow U(\bar{\mathbf{x}})$ | | | $\mathbf{x}_{3H}^2 \leftarrow U(\bar{\mathbf{x}})$ | $\cdots$ |

Table 2: Simple load balancing. The faster worker can advance both sequences, even when the slower worker has not yet finished the computation. In the example each worker does $H$ steps of local SGD (denoted by the operator $U: \mathbb{R}^d \to \mathbb{R}^d$) before writing back the updates to the aggregate $\bar{\mathbf{x}}$. Due to the load balancing, $\tau \leq 3H$.

## D    DETAILS ON EXPERIMENTS

We here state the precise procedure that was used to generate the figures in this report. As briefly stated in Section 4 we examine empirically the speedup on a logistic regression problem, $f(\mathbf{x}) = \frac{1}{n}\sum_{i=1}^n \log(1 + \exp(-b_i \mathbf{a}_i^\top \mathbf{x})) + \frac{\lambda}{2}\|\mathbf{x}\|^2$, where $\mathbf{a}_i \in \mathbb{R}^d$ and $b_i \in \{-1, +1\}$ are the data samples. The regularization parameter is set to $\lambda = 1/n$. We consider the small scale w8a dataset (Platt, 1999) ($d = 300, n = 49749$).

For each run, we initialize $\mathbf{x}_0 = \mathbf{0}_d$ and measure the number of iterations[6] (and number of stochastic gradient evaluations) to reach the target accuracy $\epsilon \in \{0.005, 0.0001\}$. As we prove convergence only for a special weighted sum of the iterates in Theorem 2.2 and not for standard criteria (last iterate or uniform average), we evaluate the function value for different weighted averages $\mathbf{y}_t = \frac{1}{\sum_{i=0}^t w_i}\sum_{i=0}^t w_i \mathbf{x}_t$, and consider the accuracy reached when one of the averages satisfies $f(\mathbf{y}_t) - f^\star \leq \epsilon$, with $f^\star := 0.126433176216545$ (numerically determined). The precise formulas for the averages that we used are given in Table 1.

For each configuration $(K, H, b, \epsilon)$, we report the best result found with any of the following two stepsizes: $\eta_t := \min(32, \frac{cn}{t+1})$ and $\eta_t = 32c$. Here $c$ is a parameter that can take the values $c = 2^i$ for $i \in \mathbb{Z}$. For each stepsize we determine the best parameter $c$ by a grid search, and consider parameter $c$ optimal, if parameters $\{2^{-2}c, 2^{-1}c, 2c, 2^2c\}$ yield worse results (i.e. more iterations to reach the target accuracy).

---

[6]Note, that besides the randomness involved the stochastic gradient computations, the averaging steps of synchronous local SGD are deterministic. Hence, these results (convergence in terms if numbers of iterations) can be reproduced by just simulating local SGD by using virtual workers (which we did for large number of $K$). For completeness, we report that all experiments were run on an an Ubuntu 16.04 machine with a 24 cores processor Intel® Xeon® CPU E5-2680 v3 @ 2.50GHz.

In Figures 4 and 5 we give additional results for mini-batch sizes $b \in \{1, 16\}$.

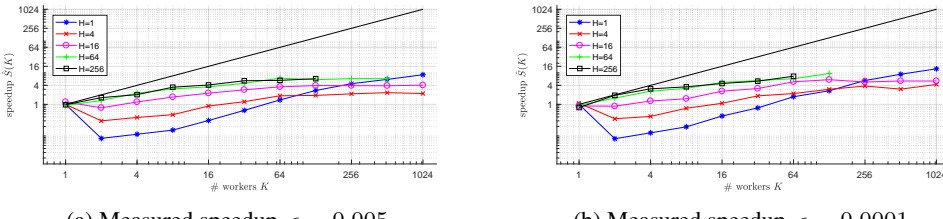

(a) Measured speedup, $\epsilon = 0.005$.

(b) Measured speedup, $\epsilon = 0.0001$.

Figure 4: Measured speedup of local SGD with mini-batch $b = 1$ for different numbers of workers $K$ and parameters $H$.

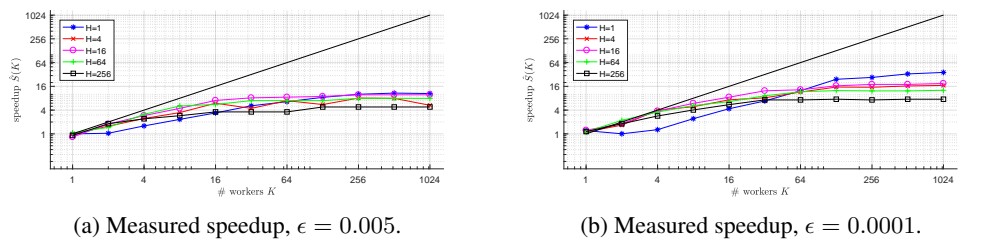

(a) Measured speedup, $\epsilon = 0.005$.

(b) Measured speedup, $\epsilon = 0.0001$.

Figure 5: Measured speedup of local SGD with mini-batch $b = 16$ for different numbers of workers $K$ and parameters $H$.

