# OpenReview forum: "Local SGD Converges Fast and Communicates Little"
_ICLR.cc/2019/Conference_

### Official Review · AnonReviewer1 · 2018-11-02
**Review: Local SGD converges fast and communicates little**

**Rating:** 8
**Confidence:** 4

**Review:**

The authors analyze the local SGD algorithm, where $K$ parallel chains of SGD are run, and the iterates are occasionally synchronized across machines by averaging. For sufficiently short intervals between synchronization, the algorithm achieves the same convergence rate in terms of the number of gradient evaluations as parallel minibatch SGD, but with the advantage that communication can be significantly reduced.

The algorithm is simple and practical, and the analysis is concise and seems like it could be applicable more generally to other parallel SGD variants.

I am curious about what happens for the analysis of the algorithm when $H$ becomes large. As the authors point out, when $H=T$, this is one-shot averaging which is known to converge. The authors mention not working too hard to optimize the bounds for extreme values of $H$, which is fine, but I wonder if this is possible using their analysis technique, or whether new tools would be necessary.

---

> ### Author Response · Authors · 2018-11-19
> **Response to Reviewer 1 - recovering the rates for one-shot averaging requires modifications to the proof**
>
> Dear Reviewer. We thank you for your time and effort that you invested into reviewing our manuscript and your favorable assessment of our work.
>
> Indeed, when H -> T, we do not recover the results for one shot averaging. In our proof we leverage the fact that less frequent averaging does not hurt the convergence as long as the local sequences are close (Lemma 3.3). However, when H=T, then Lemma 3.3 is not tight enough. Perhaps it is possible to trade-off the error introduced in Lemma 3.3 for a slower rate (e.g. only sublinear speed up), but right now we do not see how this could be achieved without significant changes to the proof.

---

> > ### Comment · AnonReviewer1 · 2018-11-29
> > **Re: Response**
> >
> > Thank you for your reply. I see no issue with the analysis failing to account for large $H$, it obviously would be nice if it could but alas it does not.
> >
> > I think that this is a well-written paper with an interesting, novel contribution to parallel optimization, and I stand by my opinion that the paper should be accepted to ICLR.

---

### Official Review · AnonReviewer3 · 2018-11-02
**Interesting direction**

**Rating:** 5
**Confidence:** 5

**Review:**

This paper presents an analysis of "local SGD", which averages estimators obtained by running SGD in separate machines once in a while. The paper presents bounds on "how frequent" the estimators required to be averaged in order to yield linear parallelization speedups. This is an interesting paper, but I have some concerns that I will elaborate on below:

[1] This paper's assumption of bounded variance of Stochastic Gradients and drawing conclusions about frequency of averaging does not reflect practical implementations of SGD for Machine Learning contexts. For example, note that in this oracle model, there exists bound on batch size (T^alpha, alpha\in[1/3,1/2]) that yield linear parallelization speedups (for example, see Dekel et al. (2012)); however, as Dekel et al (2012) note, such bounds are fairly crude estimates on a per-problem basis for practical purposes. These issues naturally continue to exist with regards to the upperbound on the frequency of communication as argued by this paper.

[2] Furthermore, the claim that such a bound on frequency of communication for local SGD which is not known before is not really true. In the convex case, the paper of Jain et al. (2016) presents a precise characterization of when to average of iterates across machines to obtain linear parallelization speedups, and this is a problem dependent quantity that works without assumptions such as bounded variance of stochastic gradients for the least squares problem. Note that, as reflective in practice, this result conveys that averaging the solutions of multiple independent runs of SGD does not help anything when the bias (initial error) dominates the variance.

[3] Note that local SGD has been known for a while and is referred to as Iterative Parameter Mixing in the literature. An example of this is the thesis of Greg Coppola (2015). A more careful literature search can provide more references/results on this topic.

[4] This paper claims that (in page 2) in order to "improve computation versus communication tradeoff, one can increase the batch size or increase communication interval". This appears to be an imprecise statement. For example, if I kept increasing batchsize without any limit, and the bias in my problem is much larger than the variance (where bias and variance follows definitions from Bach and Moulines (2011,2013)), this does not lead to any parallelization speedup. This is in contrast to when the variance dominates the bias, wherein, model averaging/increasing batch size helps. What is the reason for the authors to conclude that increasing batch size is equivalent to increasing communication interval?

---

> ### Author Response · Authors · 2018-11-19
> **Response to Reviewer 3 - our result captures the bias/variance tradeoff; but we will be more precise with the simplified statements in the introduction**
>
> Dear Reviewer. We thank you for your time and effort that you invested into reviewing our manuscript and your thoughtful comments. We hope that our answers will settle your concerns.
>
> 1) We agree with the reviewer on this comment. The bounds on H derived here have a somewhat theoretical flavor, like the asymptotic bounds derived in (Dekel et al., 2012). Our results show that H=O(sqrt(T)) is asymptotically optimal, however, as the “O” notation hides (problem specific) constants, this might not be the best choice for practical purposes, similar as discussed for the batch size in (Dekel et al., 2012).
>
> 2) We like to note a few fundamental differences from (Jain et al, 2016, https://arxiv.org/pdf/1610.03774v4.pdf ).
> (Jain et al. 2016) consider the stochastic approximation problem of Least Squares Regression, under strong convexity and bounded fourth moment assumption (we consider general strongly convex functions, but with more restrictive bounded second moment assumption). They provide analysis for SGD with constant stepsize (we consider decreasing stepsizes) for mini-batch SGD and SGD with tail-averaging. Theorem 6 discusses the averaging of *independent* runs of SGD with tail averaging, however the averaging only happens at the end (i.e. H=T, one-shot averaging). In local SGD the sequences are averaged more often, and after averaging, the sequences become correlated. This algorithm is not addressed in (Jain et al.).
>
>  (Jain et al.) show that averaging the solutions of multiple independent runs of SGD does not help when the bias (initial error) dominates the variance. Besides the already stated differences of the algorithm in (Jain et al.) and local SGD, we like to remark that a similar observation can be made in our case: In Corollary 2.3 only the variance terms enjoy a linear speedup, whereas the bias terms do not. We will add a remark on this observation and will be more careful when stating the results (see also 4) below).
>
> 3) (Coppola, 2015) show that Iterative Parameter Mixing (IPM) converges, but no speedup from parallelization has been shown (cf. pg. 94, Coppola, 2015, “In fact, a O(M) penalty is occurred”, where M=H in our notation). We like to thank the reviewer for pointing us to this reference, and to IPM in general (McDonald et al. (2010)). We will update the related work section with those appearances of “local SGD” in the literature.
>
> 4) We agree with the reviewer, that “one can increase the batch size […] to improve the computation versus communication tradeoff” is an imprecise statement and does only hold when increasing the batch size gives faster convergence (i.e. when variance dominates the bias). When we claim “one can increase the batch size or increase communication interval […] to improve the computation versus communication tradeoff” we mean the following:
>
> Clearly, [Increasing the batch size/decreasing the communication frequency] results in less communication *per iteration*. However, both strategies can also *reduce the total communication* when the variance dominates the bias term (cf. the Theorem, and Corollary 2.3). This was known for mini-batch SGD, and we prove this fact for local SGD in this paper. We will clarify the statement on page 2.

---

> > ### Comment · AnonReviewer3 · 2018-11-30
> > **Response**
> >
> > Thank you for the clarifications.
> >
> > In a sense, the works of Dekel et al , Jain et al are highly relevant references for various reasons (below) that are missed in the latest version of the paper. In particular,
> >
> > [1] Note that:
> >
> > (a) this paper's result on local SGD is a weaker upperbound relying on the bounded variance of stochastic gradients assumption (which means that iterates lie in compact set necessitating projections of iterates and so on and so forth - this doesn't reflect practice where projections of SGD's iterates are rarely carried out).
> >
> > (b) As acknowledged by the authors, note that the work of Dekel et al. (2012) is related to this effort because they presents (potentially the first to the best of my knowledge) analysis of mini-batch SGD, which is one of the ends of the local SGD framework. The work of Dekel et al (2012) also encourages future work (specifically for relative newcomers in the area) by mentioning certain weaknesses in their results.
> >
> >
> > [2] Again, despite working for the least squares case, Jain et al (2016)'s work is highly relevant to this work because:
> > (a) they present highly precise results that cover two ends of the local SGD spectrum - mini-batching and model averaging. They also consider batch-size doubling, which indicates that more machines are used at latter stages of the optimization when variance>> bias.
> > (b) as acknowledged by the authors, Jain et al (2016) work with significantly weaker assumptions than this paper. In particular, Jain et al rely on fourth moment properties of the input, do not require bounded variance assumption and present bounds applicable on a per-problem basis.
> > (c) Jain et al present precise results on the degree of parallelization permitted at different points of time in the optimization - i.e. both when bias >> variance and variance >> bias. In particular, in addition to the situation when variance dominates bias, the results of Jain et al indicate that the bias is not hurt (at the start of optimization) through mini-batching if the batch size is set appropriately (instead of being set to some arbitrarily large value).
> >
> > I would be willing to know if the authors have any issues with the points relating these works to their paper of local SGD. If not, I would suggest that the authors refer to these papers and write out limitations with their bounds and refer to the strength of certain results (for specific cases). This is to ensure that (a) there are two ways to analyze SGD and related methods (i.e. with/without bounded variance assumptions), and (b) there is future work needed to close the gap with regards to offering a precise analysis of local SGD's precise behavior.

---

> > > ### Author Response · Authors · 2018-12-02
> > > **We will address these concerns in the revision**
> > >
> > > Please excuse our negligence of not having included these references in the current revision. We agree that both algorithms work at “the end of the local SGD spectrum” (H=1 for mini-batch SGD (Deckel et al.), and H=T for the model averaging discussed in (Jain et al.)) and thus merit discussion.
> > >
> > > We will certainly address the concerns (a) [bounded gradient assumption] and (b) [future work]. We agree that future work must deepen the understanding of local SGD further. An important direction could be development of tighter bounds, of the same flavor than the ones that were obtained for the algorithms discussed in (Jain et al.), i.e. considering also bias and variance at different stages of the optimization process.
> > >
> > > However, as acknowledged by the reviewer, (Jain et al.) consider a less general function class (quadratic functions) and algorithms which are in general different from local SGD (in case H<T). Thus, we don’t see why we should mention (as suggested in [1](a)) that our results are “weaker” than the results in (Jain et al.). Perhaps the reviewer meant the limitations of the bounded gradient assumption? We will include a discussion of this point as indicated above (addressing (a)).

---

### Official Review · AnonReviewer2 · 2018-11-05
**A convergence proof for local SGD is provided. Local SGD (averaging local SGD models, once in a while) can provably provide the same speedup gains as minibatch, but may be able to communicate significantly less.**

**Rating:** 8
**Confidence:** 5

**Review:**

The authors of this paper analyze a well known technique for parallel training, where each compute node locally trains a model with SGD, and once in a while the K compute nodes average their models. Local SGD, although not as widely used as mini-batch SGD, can provide some gains in terms of the cost of communication. This can be achieved by decreasing the frequency of synchronization, while locally also increasing the minibatch.

To the best of my knowledge, the authors are the first to provide a complete theoretical analysis of local SGD for strongly convex functions. They prove that under strong convexity, and the bounded gradients assumption, local SGD will (in the worst case) achieve a linear speedup over vanilla SGD, as long as the parallel models are averaged frequently enough. They show that although frequent averaging is important for speedup, the overall communication cost can be lower than minibatch SGD that may require smaller batches and hence more frequent communication.

The authors extend their results to the asynchronous case, where a similar convergence bound is derived. The overall theory seems to be partly inspired by the perturbed iterates framework of Mania et al., however the application is novel and interesting.

The authors include some limited experimental results that validate their bounds.

This is a well-written paper, that will certainly be of interest to researchers working on stochastic optimization, and distributed learning. The results are interesting and clearly stated. The proofs seem complete and correct, and are easy to follow.

I have two minor comments:
1) In a recent paper, Dong et al. [1] suggest that for any problem (convex or nonconvex), the largest possible batch size in minibatch SGD that allows for linear speedups will be proportional to “gradient diversity”, i.e., a measure of similarity between the concurrently processed gradients. For example, when all gradient are identical, there is no speedup to be extracted. This diversity term does not seem to appear in the main theorem, as one may expect. For example, the presented bounds still seem to provide speedup gains for the case where all individual n functions are identical (eg minimum grad. diversity). This should not be possible, as there are no parallel speedups to be extracted in this case. I’m wondering how that fact is reflected in the presented bounds (maybe it’s one of the extreme parameter cases that are not covered by the main theorem).

2) The authors do not provide details of their experimental setup. For example it would be useful to know what hardware they implemented their algorithms on. It seems that they run experiments for up to 1K workers. Are these individual cores, or was this the result of hyper-threading? Finally, it’s unclear if Fig 1 is a theoretical, or an experimental curve.



[1] http://proceedings.mlr.press/v84/yin18a/yin18a.pdf

---

> ### Author Response · Authors · 2018-11-19
> **Response to Reviewer 2 - we did not consider gradient diversity as we provide data-independent bounds**
>
> Dear Reviewer. We thank you for your time and effort that you invested into reviewing our manuscript and your favorable assessment of our work. Thanks for pointing us to (Yin et al.), we will add short remark on gradient diversity in the revision (see answer to 1)). We think that decreasing the communication frequency in local SGD is especially helpful (to save communication), when the alternative strategy of increasing the batch size does not give a linear speedup (due to low gradient diversity).
>
> Concerning your minor comments:
> 1) Let us consider mini-batch SGD as an example: a *data-independent* worst case analysis, assuming just a bound on the variance on each sample of the stochastic gradient, predicts a linear speed up with respect to batch size. However, in practice, the variance of a single sample (or a batch) could be much lower than predicted by these bounds. Gradient diversity is a *data-dependent” quantity that measures this discrepancy and explains why mini-batch SGD does only enjoy linear speedup for small batch sizes for if the gradient diversity is small (Yin et al.).
>
> Our analysis is *data-independent*, that is why gradient diversity does not appear in our bounds. However, we note that the variance term explicitly appears in our Theorem and allows to extract more fine-grained results for special cases. For instance, the special case mentioned by the reviewer (all functions identical, so no speed-up possible), implies \sigma = 0, and Corollary 2.3 shows convergence at rate 1/T^2 (which is better than 1/T for the general case). However, for this example a rate of e^(-T) could be obtained with a *constant* stepsize. Our proof only covers decreasing stepsize, thus the suboptimal result is expected.
>
> 2) All experiments are conducted for the synchronous versions of the algorithm, for more details on the protocol see also Section D. Besides the randomness used to pick the stochastic gradients for each thread/local sequence, the aggregation of the local sequences after H steps is deterministic. Thus, we (as stated) simulate the number of workers by running the corresponding threads in sequence. This does not change the output of the algorithm. We will make this more precise. Figure 1 shows the theoretical expected behavior, Figure 3 the measured behavior.

---

### Public Comment · (anonymous) · 2018-11-08
**Several References are missed**

Since you are talking about local SGD, or model averaging, these two papers should be cited:
"Improving deep neural network acoustic models using generalized maxout networks" The training method in this paper is exactly same with Local SGD

and this paper
"Scalable Training of Deep Learning Machines by Incremental Block Training with Intra-block Parallel Optimization and Blockwise Model-Update Filtering" This paper propose to build connections between successive synchronizations of "local sgd"

---

### Meta-Review · Area_Chair1 · 2018-12-13
**Good analysis of local SGD, but still needs discussion of closely related prior work**

**Confidence:** 4
**Recommendation:** Accept (Poster)

**Metareview:**

This paper analyzes local SGD optimization for strongly convex functions, and proves that local SGD enjoys a linear speedup (in the number of workers and minibatch size) over vanilla SGD, while also communicating less than distributed mini-batch SGD. A similar analysis is also provided for the asynchronous case, and limited empirical confirmation of the theory is provided. The main weakness of the current revision is that it does not yet properly relate this work to two prior publications: Dekel et al., 2012 (https://arxiv.org/pdf/1012.1367.pdf) and Jain et al., 2016 (https://arxiv.org/abs/1610.03774). It is critical that these references and suitable discussion be added in the camera-ready paper, since this issue was the subject of considerable discussion and the authors promised to include the references and discussion in the final paper.